

# Ideas and perspectives: Tree-atmosphere interaction responds to water-related stem variations

Tim van Emmerik[1], Susan Steele-Dunne[1], Pierre Gentine[2], Rafael S. Oliveira[3], Paulo Bittencourt[3], Fernanda Barros[3], and Nick van de Giesen[1]

[1]Water Resources Section, Delft University of Technology, The Netherlands.
[2]Department of Earth and Environmental Engineering, Columbia University, New York, USA.
[3]Department of Plant Biology, Institute of Biology, University of Campinas, Campinas, SP, Brazil

**Correspondence:** Tim van Emmerik (t.h.m.vanemmerik@tudelft.nl)

**Abstract.** Land-atmosphere interactions depend on momentum transfer from the atmosphere to the canopy, which in turn depends on the tree drag coefficient. It is known that the drag coefficient, and thus tree-atmosphere momentum transfer, can vary strongly within a canopy. Yet, only few measurements are available to study the variation of tree-atmosphere momentum transfer in time and space, and in response to tree water deficit. In this paper we use accelerometers to estimate tree-atmosphere
momentum transfer for 19 individual trees of seven different species in the Brazilian Amazon. The five-month measurement period included the transition from wet to dry months. Here, we demonstrate that under field conditions, tree-atmosphere momentum transfer can vary considerably in time and space. Increased water-related stem variations during the dry months is related to observed changes in tree-atmosphere momentum transfer, which is hypothesized to be caused by tree water deficit induced changes in tree mass.

**1  Introduction**

The atmospheric boundary layer and the land surface directly influence each other through momentum, mass and energy exchange (Gentine et al., 2011, 2012; Green et al., 2017). Land-atmosphere interactions influence meteorological and hydrological fluxes and states, and biotic and abiotic processes, such as seed and pollen distribution (Katul et al., 2005), deposition of atmospheric pollutants (Clifton et al., 2017), and transfer of water, heat and $CO_2$ (Aumond et al., 2013). Land-atmosphere
interactions are influenced by the momentum transfer from atmosphere to the canopy, which highly depends on the turbulent drag coefficient of individual trees (Poggi and Katul, 2007). Drag causes loss of momentum, and the interplay between canopy and atmosphere is heavily affected by the transport of water, heat, and carbon between vegetation and the atmosphere (Molina-Aiz et al., 2006; Cescatti and Marcolla, 2004). In meteorological models, large-eddy simulations, and land-surface models, tree-atmosphere momentum transfer plays a crucial role in describing energy and momentum transfer from the atmosphere
to the canopy. Such transfer between trees and atmosphere greatly depends on the tree drag coefficient (Gillies et al., 2002). For computational ease, the drag coefficient is often assumed constant (Katul et al., 2006; Cassiani et al., 2008; Dupont and Brunet, 2008), both in time and space. It is known, however, that the drag coefficent, and thus the degree of tree-atmosphere





momentum transfer, can vary strongly within a canopy, and as a function of environmental conditions (Belcher et al., 2012). Assuming a constant drag coefficient may therefore be unrealistic, and introduces a large source of error.

Misrepresentation of the variability in canopy drag is largely due to a lack of (field) data. Various studies have quantified canopy drag (coefficients) in laboratory and field setups (Mayhead, 1973; Koizumi et al., 2010). Widely used drag coefficients
for several tree species originate from a wind tunnel experiment by Mayhead (1973). Here it was found that drag coefficient is variable between individual trees, and strongly depends on wind speed. Most wind tunnel studies used dwarf species, juvenile crowns, or miniature model trees or forests, which are not representative for actual-sized trees under natural conditions (Johnson et al., 1982; Rudnicki et al., 2004; Vollsinger et al., 2005; Meroney, 1968; Novak et al., 2000; Guan et al., 2003; Poggi and Katul, 2007). Recent work (Koizumi et al., 2010, 2016) presented a novel field method that can measure stem deflection, which
is used to derive the tree drag coefficient. However, this method requires a considerable amount of power, making it difficult to obtain long time series. This is especially a problem under field conditions, where power supply is limited.

A recently presented measurement technique (van Emmerik et al., 2017a) used low-cost accelerometers to measure tree sway. Tree sway is a result of momentum transfer from the atmosphere to the tree, and can therefore be used to study tree-atmosphere momentum transfer. The robustness of the sensors allows deployment in harsh conditions such as tropical environments, to
obtain long time series. This paper uses tree sway measurements obtained during a five month period to quantify and compare the tree-canopy momentum transfer for 19 trees in the Brazilian Amazon. The measurement period includes the transition from wet to dry months. We derived a measure for canopy-atmosphere momentum transfer, showing clear variation between species, over time, and in response to water-related stem variations.

The Amazon contains half of the world's rainforests. Yet, it remains a poorly understood component of the global carbon
and water cycle (Saatchi et al., 2007; Binks et al., 2016; Anber et al., 2015). For example, the extensively studied 2005 drought reversed the Amazon from a long-term carbon sink into a carbon source (Phillips et al., 2009). Amazon forests appear to be sensitive to increasing moisture stress (van Emmerik et al., 2017b), and future droughts have the potential to considerably change the water and carbon balance (and thus climate change) (Phillips et al., 2009). Improved understanding of the variation and dynamics of the drag coefficient will therefore contribute to a better understanding of the Amazon's role in the water and
carbon cycle, and its response to droughts.

A long time series of tree acceleration data was used to investigate tree-atmosphere momentum transfer under field conditions, and in response to increased water-related stem variations. Using an experimental method, We aim to show that tree-atmosphere momentum transfer various more than often assumed. Specifically, we demonstrate the effect of increased water-related stem variations for various tree species and individuals, which is hypothesized to be caused by water deficit
induced mass changes in the trees.



## 2 Methods

### 2.1 Study area

The field measurements of this study were obtained from August, 2015 to January, 2016 at the K34 research station in the Amazon rainforest (2.6085° S, 60.2093° W), 60 km Northwest of Manaus, Brazil. The study area is characterized by a tropical

5 monsoon climate with an average dry season from July to September (De Gonçalves et al., 2013). During the measurement period there was about 12 hours of daylight, roughly between 6 A.M. and 6 P.M. local time. Meteorological data was measured at a flux tower on site. Wind speed, temperature, and precipitation were measured every 15 minutes. For this study, we use data from the period August, 2015 to January, 2016.

### 2.2 Plant material

10 A total of 19 individual trees were measured during this experiment, covering seven tree species, and a broad range of average height and wood density. In total 7 species were measured, with 1 to 3 individuals per species. Trees were selected to cover a broad range of heights ($h$), widths (diameter at breast height, $D_{BH}$), and wood densities ($\rho_w$). An overview of the measured trees is found in Table 1.

**Table 1.** Tree characteristics: Tree number, scientific name, wood density, estimated total height and diameter at breast height ($D_{BH}$).

| Tree no. | Name | Estimated wood density $[10^3 kg/m^3]$ | Estimated height [m] | $D_{BH}$ [cm] |
|---|---|---|---|---|
| 1 - 3 | *Goupia glabra* | 0.7 (High) | 25 - 32 | 135.0 - 242.5 |
| 4 - 6 | *Lecythis prancei* | 0.875 (Intermediate) | 24 - 35 | 108.4 - 116.5 |
| 7 - 9 | *Scleronema micranthum* | 0.5 - 0.7 (Low) | 26 - 38 | 81.0 - 189.5 |
| 10 - 13 | *Eschweilera coriacea* | 0.8 (Intermediate) | 18 - 27 | 92.4 - 268.0 |
| 14 - 15 | *Dypterix odorata* | 1.1 (High) | 32 - 35 | 177.0 - 219.5 |
| 16 - 17 | *Pouteria anomala* | 0.7 (Low) | 22 - 23 | 111.0 - 117.5 |
| 18 - 20 | *Maquira sclerophylla* | 0.5 (Low) | 18 - 35 | 90.6 - 264.0 |

Wood density values were taken from the Global Wood Density Database (Zanne et al., 2009). Total tree height was measured

15 using measurement tape. Tree species were determined by a classified taxonomist. Diameter at breast height (DBH) was measured using measuring tape on the day of installation of the accelerometers. Aboveground biomass (AGB) was estimated for every tree using the pantropical model introduced by Chave et al. (2014). In this model, tree height $h$, diameter at breast height $D_{BH}$, and wood density $\rho_w$ are related to AGB through the following equation:

$$AGB = 0.0673(\rho_w \cdot DBH^2 \cdot h) \tag{1}$$



## 2.3 Experimental setup

Water proof, robust accelerometers (Acceleration Logger - Model AL100, Oregon Research Electronics, Tangent, OR, USA) were used to measure three-dimensional acceleration with a frequency of 10 Hz. The accelerometers were placed directly below the main branching of the tree, to measure the largest signal can be measured, and to minimize effect of oscillations from

primary and secondary branches (Spatz and Theckes, 2013). The frequency spectrum of horizontal, single axis acceleration was determined using a sliding window fast Fourier transform. The spectrum was estimated every 10 minutes, using a 30 minute window. In the following analysis, we use the logarithmic slope [dB/Hz] of the tree acceleration and wind frequency spectrum. The slope of the spectrum represents the damping of the driving wind force and can be seen as a measure of momentum transfer. As the tree movement is driven by wind, a part of the wind energy is transferred to kinetic energy in the tree. The

intensity of the transfer depends on the wind speed and on the tree characteristics (such as moment of inertia, mass, and the drag coefficient). For this study, the slope of the frequency spectrum between 0.2 and 1 Hz was determined every 10 min. More detailed information on the accelerometer can be found in (van Emmerik et al., 2017a, 2018, in review, 2018).

Dendrometers (ZN12-T-2IP, Natkon.ch, Switzerland) were installed at 1.5 meters above ground level. Bark thickness was measured every 10 minutes. Bark time series were used as a direct measure of water-related stem variations in trees (Zweifel

et al., 2005; Ehrenberger et al., 2012). First, the local maximum values are connected. The resulting line is the growth line, which represents the maximum stem radius in case of no water limitations. The difference between the growth line and the actual bark thickness is then the water-related stem variations, using:

$$\Delta W = D_{b,pot} - D_{b,act} \tag{2}$$

With total water-related stem variation $\Delta W$, growth line $D_{b,pot}$, and change in bark thickness $D_{b,act}$. Fig. 1 illustrates how

the growth line and water-related stem variation can be obtained from a dendrometer dataset.

## 2.4 Relating wind to tree motion

The relation between the input wind energy spectrum $P_u$ [dB], and the output energy spectrum of tree motion $P_y$ [dB], both as function of frequency $f$ can be described as:

$$P_y(f) = |H(f)|^2 \rho_a^2 C_D^2 A^2 \bar{u}^2 H_a(f)^2 P_u(f) \tag{3}$$

with mechanical transfer function $H$ [s/kg], air density $\rho_a$ [kg/m$^3$], drag coefficient $C_d$ [-], tree catch area $A$ [m$^2$], mean wind speed $\bar{u}$ [m/s], aerodynamic transfer function $H_a(f)$ [-], and power spectrum of the wind $P_u(f)$ [dB]. In general, the aerodynamic transfer function can be approximated as $H_a(f)^2 = 1$, as there is a minimal turbulent storage term (Amtmann, 1985; Mayer, 1987).

The energy conservation and dissipation of wind turbulence depends on the scale. Large scale eddies (low frequencies) are

energy containing, whereas energy dissipation mainly happens at smaller molecular scales (higher frequencies). The range in





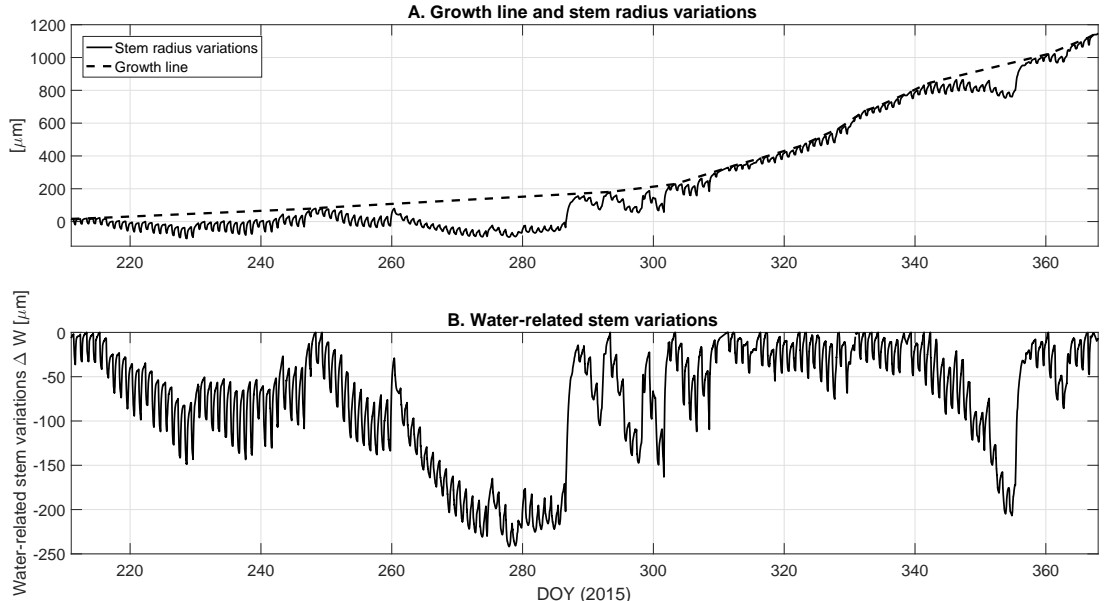

**Figure 1.** A. Growth line and steam radius variation for *Goupia glabra* tree 1, and B. the calculated water-related stem variation $\Delta W$ [$\mu$ m], from August 2015 to December 2015.

between, the inertial subrange, is where energy is transferred from low to high frequencies. Kolmogorov (1941) hypothesized that at high Reynolds numbers and under homogeneous and isotopic turbulence the inertial subrange would follow a -5/3 spectrum. The wind energy content $P_u$ within the inertial subrange is a universal function of the frequency, and can therefore be expressed as (Stull, 2012, p. 390–391):

$$P_y = C\epsilon^{\frac{2}{3}} f^{\frac{-5}{3}} = A f^{\frac{-5}{3}} \tag{4}$$

With constant $C$, dissipation rate $\epsilon$, and frequency $f$. Wind in forest canopies also exhibit this spectrum (e.g. (Flesch and Wilson, 1999; Odijk, 2015)). For turbulent conditions, the input wind spectra (see eq. 4) and its slope are known. Comparing the input Kolmogorov wind spectrum with the output tree acceleration spectrum therefore gives a measure of the momentum damping/absorption by the tree. For eqs. 4 and 5, a constant value for the wind spectrum slope (-5/3) is used in subsequent analyses. With known acceleration and wind spectra, we can find an expression for the transfer function, which is a measure of tree-atmosphere momentum transfer. The slope of the acceleration and wind spectra are related through the following equation:

$$\frac{dP_y}{df} = H^2 \rho^2 A^2 C_d^2 u^2 \frac{dP_u}{df} \tag{5}$$





With tree acceleration spectrum slope $\frac{dP_y}{df}$ [dB/Hz] and wind spectrum slope $\frac{dP_u}{df}$ [dB/Hz]. We simplify the slopes of the acceleration and wind spectra to:

$$s_a = \frac{dP_y}{df} \tag{6}$$

$$s_w = \frac{dP_u}{df} \tag{7}$$

By combining the constant variables we can simplify this to:

$$\alpha = H\rho A \tag{8}$$

$$s_a = \alpha^2 C_d^2 u^2 s_w \tag{9}$$

$$\alpha C_d = \sqrt{\frac{\frac{s_a}{s_w}}{u^2}} \tag{10}$$

with acceleration and wind spectra slopes $s_a$ and $s_w$ [dB/Hz], and transfer parameter $\alpha$ [s/m]. The combined term $\alpha C_d$ is

used as a conceptual expression for tree-atmosphere momentum transfer, and accounts for the combined effect of e.g., drag coefficient, mass, density, wind catch area.

Interaction between wind and a tree is a function of wind speed. As e.g. Mayhead (1973) and Koizumi et al. (2010), have shown, the drag coefficient and momentum transfer decrease with increasing wind speed. This is mainly due to streamlining of the tree, which decreases the catch area of the tree. In this study, we analyze the changes in the relation between the measure of

tree-atmosphere momentum transfer $\alpha C_d$ and wind speed. For each week, the following function is fit to the relation between wind speed $u$ and $\alpha C_d$:

$$\alpha C_d = A \cdot \exp(-\beta \cdot u) \tag{11}$$

with $A = 1$, and damping parameter $\beta$. $\beta$ determines the shape of the relation, and describes how the tree-atmosphere momentum transfer changes with wind speed. The value of $\beta$ is therefore used to track the variation in tree-atmosphere momentum

transfer over space and time. For better comparison between individual trees, $\beta$ is presented normalized by the mean value for $\beta$ per individual tree.





## 2.5 Data processing

We estimate the frequency spectrum of the horizontal, single axis acceleration using a sliding window fast Fourier transform (FFT). The spectrum was estimated every 10 minutes, using a window length of 30 minutes. The slope of the spectrum represents the damping of the driving wind force by the tree, and can be seen as a measure of energy and momentum transfer (van Emmerik et al., 2017a). As tree movement is driven by wind, a part of the wind energy is transferred to kinetic energy in the tree. For this study, the slope of the frequency spectrum between 0.2 and 1 Hz was determined, for every 10 minutes. The slope is presented on a logarithmic scale [Hz/dB].

## 3 Results and discussion

### 3.1 Acceleration spectra slope

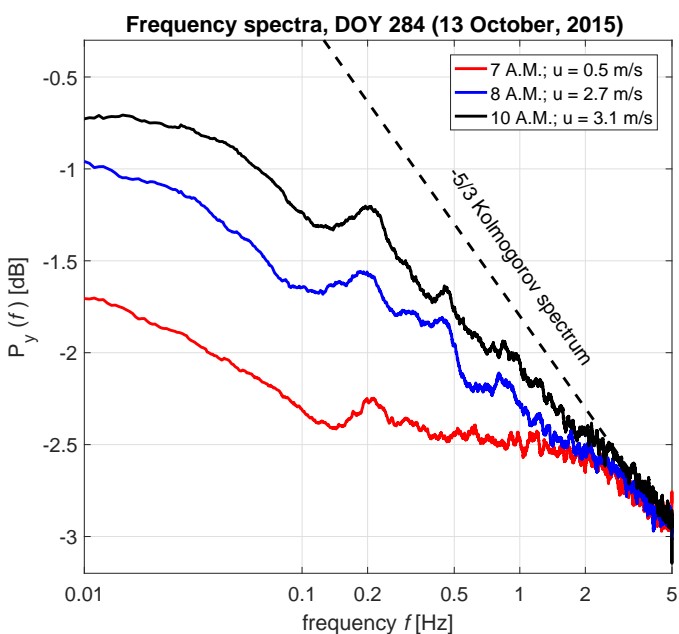

**Figure 2.** Frequency spectra of a *Goupia glabra* tree for different wind speeds on day of year (DOY) 284 (11 October, 2015), including turbulent wind spectrum (dashed black), taken from van Emmerik et al. (2017a).

Fig. 2 presents a typical acceleration energy spectrum for a *Goupia glabra* tree, for three different wind speeds. For increasing wind speeds, the slope of the spectrum approaches the Kolmogorov -5/3 spectrum. As hypothesized by Kolmogorov (1941), turbulent motions in the intertial subrange are statistically isotropic, and the wind energy spectrum is only a function of frequency. Around 0.2 Hz a peak can be observed, which is the natural frequency $f_n$ [Hz] of the tree. Variation in the natural



frequency can be related to tree mass changes, and intercepted rainfall by the canopy (van Emmerik et al., 2017a), but is not further used in this manuscript.



**Figure 3.** Tree acceleration spectra slope $s_a$ [Hz/dB] over time for DOY 301 to 306 (2015), for A. *Goupia glabra*, B. *Lecythis prancei*, C. *Scleronema micranthum*, D. *Eschweilera coriacea*, E. *Dypterix odorata*, F. *Pouteria anomala*, G. *Maquira sclerophylla*, and H. wind. Each line in A-G. represents an individual tree.

The slope varies over time, and per tree species and individual. Fig. 3 presents the acceleration spectra slope $s_a$ for each tree, grouped per tree species. As expected, the slope of the acceleration spectrum increases with wind speed. The timing and magnitude does change per tree. For example, the *Scleronema* trees (Fig. 3C) have a consistently higher slope (1.4 Hz/dB during the day) than the *Dypterix* (Fig. 3E) trees (1 Hz/dB during the day). The sum of the differences between trees are captured by the parameter $\alpha C_d$, which will be presented later. Other clear differences can be seen between individuals of





different species. Where for the *Scleronema* (Fig. 3C) and *Pouteria* (Fig. 3F) trees the slope is similar between the individuals, for *Maquira* (Fig. 3G) and *Lecythis* (Fig. 3B) the variation between the individuals is considerably larger.

## 3.2 Tree-atmosphere momentum transfer across time and space

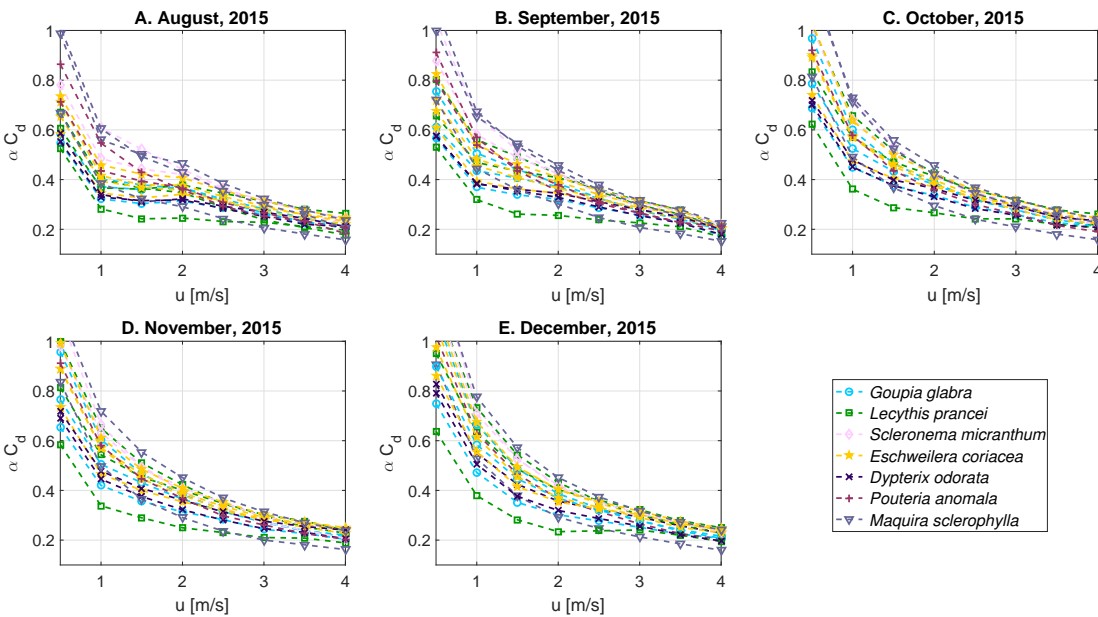

**Figure 4.** Monthly averaged tree-atmosphere momentum transfer $\alpha C_d$ per individual tree as a function of wind speed $u$ [m/s] for A. August, 2015, B. September, 2015, C. October, 2015, D. November, 2015, and E. December, 2015.

The momentum transfer between trees and the atmosphere is expressed by the $\alpha C_d$, which included effects of mass, ge-
5 ometry, wind catch area, and drag coefficient (see eq. 8). Streamlining of a tree for increasing wind speed affects the relation between $\alpha C_d$ and wind speed (Mayhead, 1973; Koizumi et al., 2010), as can be seen in Fig. 4. Here, the monthly averaged relation between $\alpha C_d$ and wind speed are presented for August to December, 2015. For wind speeds between 0 and 3-4 m/s, $\alpha C_d$ decreases, with the highest decrease between 0 and 1-1.5 m/s. For higher wind speeds $\alpha C_d$ becomes more stable. It can be seen that for $\alpha C_d$ varies considerably between individual trees, and that the relation between $\alpha C_d$ and wind speed changes
10 over time. For example, the range of $\alpha C_d$ at 1 m/s changed from 0.3-0.6 to 0.4-0.8 between August and December, 2015. It is hypothesized that this is due to changes in mass, related to e.g. water content or leaf fall.




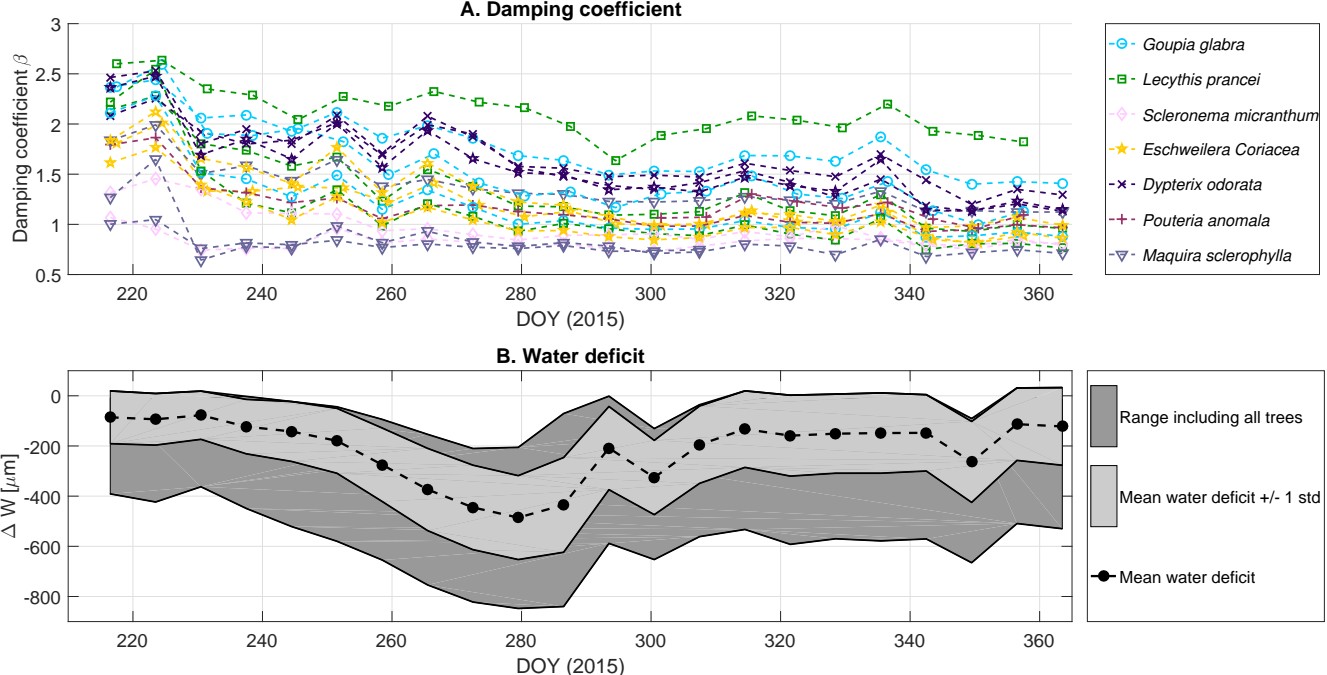

**Figure 5.** A. Normalized damping coefficient $\beta$ for each individual tree, and B. mean, mean +/- 1 standard deviation, minimum and maximum water-related stem variation $\Delta W$, for every week from DOY 218 (August) to 365 (December), 2015.

### 3.3 Effect of dry months

To further explore the changes in the relation between $\alpha C_d$ and wind, this relation was fit for each week of available data. The most important parameter is the damping coefficient $\beta$ (see eq. 9). Fig. 5 presents the weekly values for the normalized damping coefficient $\beta$ between August (DOY 220) and December (365), 2015. Recall that for the normalization, time series of $\beta$ are

5 normalized by the average value of $\beta$ for each individual tree. Water-related stem variation measurements were also available, and are also shown in Fig. 5. As most trees showed similar temporal behavior, the figure presents the average water-related stem variation based on all trees, including the minimum and maximum of the measured range.

Between DOY 230 and 280 $\beta$ decreased while water-related stem variations increased. Although water-related stem variation decreased steeply between DOY 280 and 285, $\beta$ continued decreasing until around DOY 300. Water deficit remained

10 relatively stable between DOY 285 and 340, after which a steep increase was observed between DOY 340 and 360. $\beta$ recovered between DOY 300 and 340, and decreased again between DOY 340 and 360. Changes in $\beta$ are hypothesized to be caused by changes in mass related to water content, or leaf fall. The increase in water-related stem variation supports this hypothesis, as a decreasing/increasing $\beta$ coincided with inverse changes in water-related stem variation.





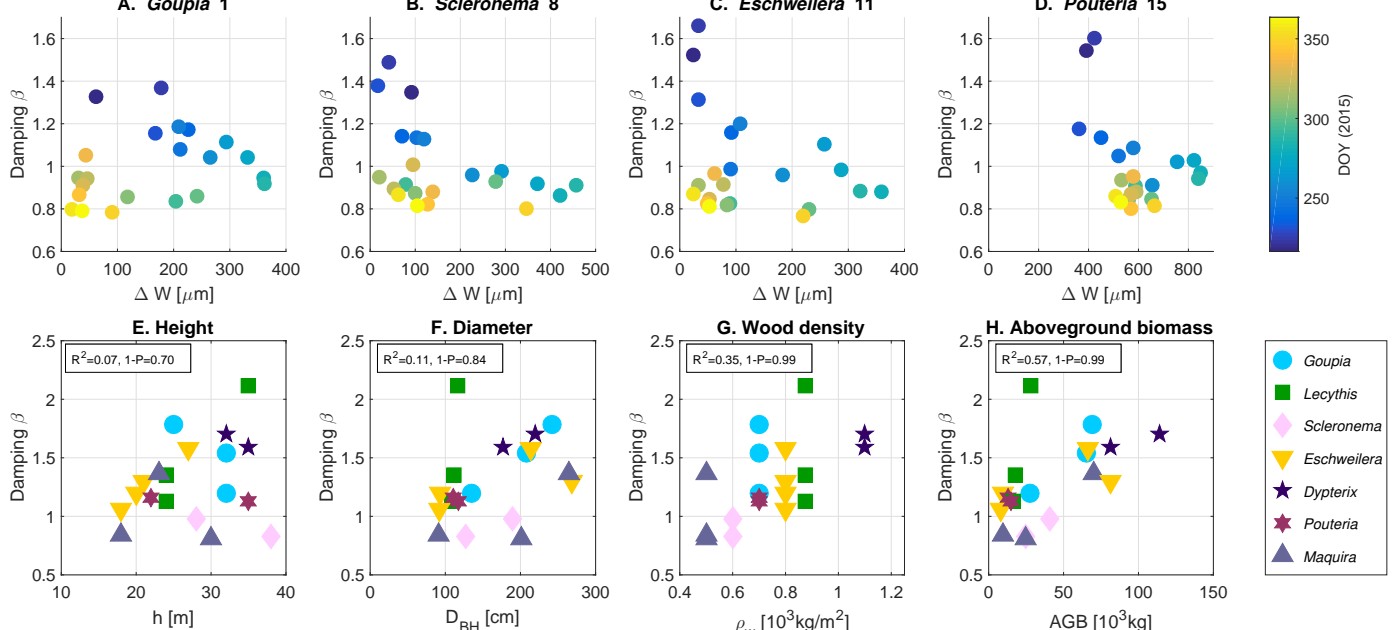

**Figure 6.** Damping coefficient $\beta$ (for fitted function $\alpha C_d$ vs. $u$) for a single A. *Goupia*, B. *Scleronema*, C. *Eschweilera*, and D. *Pouteria* tree, versus water-related stem variation $\Delta W$, colored by day of year (DOY, 2015), and average damping coefficient $\beta$ in relation to E. tree height $h$, F. diameter at breast height $D_{BH}$, G. wood density, and F. Aboveground biomass (AGB), for each measured tree.

Fig. 6 presents the relation between $\beta$, water-related stem variation and DOY for four trees. Here, it can be seen that there is a clear relation between increasing water-related stem variation and decreasing $\beta$. When water-related stem variation increases, $\beta$ drops significantly (DOY 220 to 250). For higher water-related stem variation (DOY 250 to 300), $\beta$ decreases more gradually. For the recovery (DOY 300 to 340), when water-related stem variation decreases again, the relation between $\beta$ and water-

5  related stem variation is different. Here, $\beta$ only increases gradually again for decreases water-related stem variation. For the increase in water-related stem variation between DOY 340 and 360, a drop in $\beta$ can be seen for the *Goupia*, *Scleronema*, and *Eschweilera*. Note that water-related stem variation is not a measure of water content. When water-related stem variation decreases again after DOY 300, tree water content or biomass do not necessarily increase. Therefore, it can be expected that $\beta$ does not directly increase either, resulting in the observed hysteresis pattern.

10  To explore what might explain the variation in $\beta$, Fig. 6E-H. present the average value of $\beta$ (for August to December, 2015) in relation to the estimated physical tree properties. As expected, the momentum transfer between tree and atmosphere is the sum of various different factors. It can be seen that for most species, increased height, and diameter at breast height $\beta$ is higher. The relation is weaker when looking at all trees combined ($R^2 = 0.07$ and $R^2 = 0.11$, respectively) It can also clearly be seen that for increased wood density the average $\beta$ is higher ($R^2 = 0.35$), which can be explained by the coinciding increase in





tree stiffness. We also see that there is a relation between aboveground biomass and $\beta$ for all trees, and for each tree species separately. Higher tree biomass results in higher $\beta$ ($R^2 = 0.57$). This suggests that changes in $\beta$ can be explained by water variations, due to for example leaf flush or fall and water content related biomass changes.

The relation between $\beta$ and can explain the observations in Fig. 5 and 6A-D. During periods of increased water-related stem

variation during the dry months, $\beta$ clearly decreases for each tree. The increased water-related stem variation suggests that this is due to water content related mass changes in the vegetation, such as decreasing tree water content and increased leaf fall during dry months. The relation between aboveground biomass and $\beta$ supports this hypothesis, as the observed decrease in $\beta$ might be explained by decreasing tree mass during the dry months.

### 3.4 Synthesis

The results presented in this paper show that the degree of tree-atmosphere momentum transfer varies considerably between species. Previous work has shown this for some species using wind tunnel experiments. This study uses *in situ* measurements to demonstrate the variation in tree-atmosphere momentum transfer in the field. Besides variation in space, significant temporal variation in tree-atmosphere momentum transfer was found. To our knowledge, this is the first time that this has been measured under field conditions.

Changes in tree-atmosphere momentum transfer seem to be related to changes in tree mass. For the period with increased measured water-related stem variation, $\beta$ decreased. We hypothesize that this is a direct effect of increased tree water deficit. There are two mechanisms that might explain the changes in $\beta$. First, tree mass might change through changes in tree water content. When tree water deficit is increasing, insufficient water is available to refill the storage,and tree water content decreases. In future work we recommend measurements of leaf water potential or tree water content to assess the influence of

tree mass change on tree-atmosphere interaction in more detail. Second, changes in $\beta$ might also be caused by leaf fall, which for some trees might also affect the total mass significantly (Lopes et al., 2016). Leaf fall can be a direct consequence of tree water deficit (Reich and Borchert, 1984), but is not always driven by water deficit. Peak rates of leaf fall almost always occur during the dry months in tropical forests (Wright and Cornejo, 1990). This might explain the quick response to increased water-related stem variation. When water-related stem variation is low again, one might expect a recovery in $\beta$ as well. However, if $\beta$

is mainly changed due to leaf fall, the recovery might be delayed significantly. Leaf expansion might occur only a few weeks during the early wet months (Reich and Borchert, 1984), growth of new leaves only occurs as long as soil moisture is plentiful (Bordiert, 1994). Absence of these conditions could explain the slow response in $\beta$. For this reason a hysteresis pattern is observed for damping and water-related stem variation in Fig. 6. Previous work on the K34 site showed that the highest annual litterfall occurs between August and October Wu et al. (2016), coinciding with the period of decreased $\beta$ in this study. For

increased water-related stem variation it is likely that tree water content and biomass decreased. However, during periods of low water-related stem variation trees are not restored, and it takes time before tree water content and biomass increase again. Therefore $\beta$ does also not directly increase. Additional data such as leaf water potential or eddy covariance data will help attributing the change in tree-atmosphere momentum transfer to changes in tree water content, leaf fall, or other mechanisms not yet considered.





The impact of water-related stem variation on tree-atmosphere is a significant finding, as this shows that tree-atmosphere momentum transfer is also affected during the dry months, in addition to the general spatiotemporal variation. This sheds a new light on momentum transfer from the atmosphere to the tree. Previous studies on tree-atmosphere momentum transfer used the drag coefficient $C_d$ as a measure for tree-atmosphere momentum transfer. It was found that this varies with wind speed, and per

tree species. So far, this has not been done on trees in forests. Also, no studies have investigated the effect of water-related stem variation on tree drag coefficient, or any other measure of tree-atmosphere momentum transfer. Additional high resolution wind measurements would allow further, more detailed, exploration of tree-atmosphere interaction. For example, the conceptual $\alpha$ parameter can be quantified using more detailed wind measurements, yielding a more exact and physically based expression of tree-atmosphere interaction. We found that $\alpha C_d$ varies across different time scales. We recommend further research to study

the factors that dominate the variations at different time scales, which may be important for modeling purposes focusing on atmospheric processes on different time scales.

With this paper we aim to show that assuming a constant drag coefficient is unrealistic, and potentially introduces a large source of error. For example, several large-scale land-surface models approaches represent the canopy layer, and its interaction with the atmospheric boundary layer, through static parameters. The observations presented in this paper show that future

efforts should consider using more dynamic representation and parameterization to reduce errors. Additional measurements of turbulent kinetic energy, (high frequency) wind speed at specific trees, leaf water potential and leaf area will give more insights in the dynamics driving changes in tree-atmosphere interaction, and will allow for expressing tree-atmosphere interaction in terms of actual drag coefficient.

This paper demonstrates that the variation in tree-atmosphere momentum transfer can change considerably during the shift

from the wet to the dry months. This has important implications for the water and carbon balance, as these depend strongly on the momentum transfer from atmosphere to the canopy. We emphasize that this papers presents experimental work. Experimental work is imperative to gain a better understanding of governing processes (van Emmerik et al., in press, 2018), in this case regarding tree-atmosphere interactions . However, additional measurements of the input wind spectra, and its variation over time and space, is crucial for further exploration of the relation between wind, tree sway, and momentum transfer. Combining

the current data with plant physiological measurements will allow further testing of the hypothesis that the temporal changes in tree-atmosphere momentum transfer are related to water deficit induced tree mass changes.

## 4    Conclusions

Tree sway measurements were used to estimate tree-atmosphere momentum transfer for 19 trees during a transition from the wet to the dry months in the Brazilian Amazon. It was found that tree-atmosphere momentum transfer varies considerably

between individuals and between species.

Tree-atmosphere momentum transfer, and its relation with wind speed also changes over time. Especially during the transition from the wet to the dry months, a clear change in tree-atmosphere momentum transfer was measured for all trees. The change in tree-atmosphere momentum transfer coincided with increasing water-related stem variation.





A positive relation was found between estimated aboveground tree biomass, and average tree-atmosphere momentum transfer. This suggests that the variation in accelerometer-derived measure of tree-atmosphere momentum transfer is caused by changes in tree mass, most likely caused by water tree deficit induced changes in water content or leaf fall.

Our experimental work provides new insights in variation in tree-atmosphere momentum transfer in time and space, and its response to increased water-related stem variation. We aim to show that using static parameterization of vegetation in for example land-atmosphere or climate models might introduce a source of error. Future work should focus on attributing changes in tree-atmosphere interactions to changes in tree water content, leaf fall and flush, and other mechanisms.

*Data availability.* Dendrometer data may be obtained from Fernanda de V. Barros (email: nandavascon@gmail.com). Accelerometer data may be obtained from Tim van Emmerik (email: t.h.m.vanemmerik@tudelft.nl) or can be downloaded through here 10.4121/uuid:c9974180-aa9b-40b4-8d...

*Author contributions.* TvE, SSD, PG, NvdG designed the study, TvE, RSO, PB, FB conducted the fieldwork, TvE performed the initial data analysis and write the first manuscript, all authors contributed to data interpretation and writing of the manuscript.

*Competing interests.* The authors declare no competing interests are present.

*Acknowledgements.* We thank FAPESP GOAmazon project 2013/50431-2 in whose field campaign data was collected. We thank the Large Scale Biosphere-Atmosphere (LBA) Program at the National Institute for Amazon Research (INPA) for logistical and infrastructure support during field measurements. The work of S. C. Steele-Dunne was supported by a Vidi Grant 14126 from the Dutch Technology Foundation STW, which is part of The Netherlands Organisation for Scientific Research (NWO), and which is partly funded by the Ministry of Economic Affairs. The authors are very grateful to Laura Borma.



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
