# Peer review of "Ideas and perspectives: Tree-atmosphere interaction responds to water-related stem variations"

_Biogeosciences, 2018_

## Referee Comment (RC1) · Anonymous Referee #1 · 23 Jul 2018

The manuscript "Tree-atmosphere interaction responds to water-related stem variations" brings novel data and interpretation on the dynamics of atmosphere forest interaction by applying an original methodology. In short it suggests that changes in tree mass due to lower water content relates to transfer of momentum from air masses to trees, quantified by their sway. The manuscript would benefit by a more careful structuring as is quite repetitive in several instances. It did kept me wondering to what extent does tree water content varies. Is there any data out there quantifying water volume/mass changes trough time? In percentage, how much do mass varies in a tree (1%, 5%, 25%)? In addition, is it possible that a lower water potential on the tree trunk cause changes in elasticity (flexibility) of the trunk? Furthermore, due to

decreased leaf area index during dry months, is it possible that the changes in wind patterns within canopies space (not measured in this study) produced the observed temporal patterns? Because the speculative nature of this study (conclusion depending on an inferred water content), if would be great to have a deeper discussion on other alternative hypothesis.

Page 1: Line 5: Better use monitoring than "measurement". Page 1: Line 2: Not sure what "meteorological fluxes" means. Page 1: Line 7: to "vary considerably" is quite vague. Provide a percentage (or range) of variation. Furthermore, need to specify "variations" in what (I suppose tree trunk diameter)? This also apply to the title of the manuscript. Page 2: Line 27: no need for capitalized "We". Line 28: change ". . .momentum transfer various more..." to "momentum transfer varies more" Page 3; Line 8: Can you indicate the source of the meteorological data? Page 3; Line 11: A bit repetitive (7 tree species). Page 3; Line 15: Change to "measuring tape". Page 4; Line 13: How long was the point dendrometers allowed to settle before reliable measurements could be taken. I mention that because it is said that regular band dendrometers requires some months to settle in place, preventing immediate reliable monitoring. Page 12; Line 4-5: Higher biomass do not necessarily relates to higher water content. Page 12; Line 6: "The relation between $\beta$ and can explain the observations in Fig. 5 and 6A-D." this phrase needs improvement. The observations suggest a relationship.

---

## Referee Comment (RC2) · Anonymous Referee #2 · 28 Jul 2018

This manuscript applies a novel, low cost, field-durable method to examine an oft-oversimplified process in land surface models: the momentum transfer between the atmosphere and tree canopies. Its major contributions include (1) being the first to achieve in situ measurement of tree-atmosphere momentum transfer, and (2) finding significant fine-scale spatial variability (individual tree) and coarser-scale temporal variability (over weeks and seasons) in tree-atmosphere momentum transfer. Considering the high spatiotemporal variability observed, it would have been nice to see some back-of-the-envelope estimate of its relevance to related processes, like those mentioned in the introduction (i.e., transfer of water, heat or gasses). But, such an estimate is not necessary. This method, dataset and analysis would be of high interest to the geoscience community with some revision:

*Why does the title and abstract focus solely on "water-related stem variations"? The authors discuss the potential effect of leaf mass variations late in the manuscript - leaves are not mentioned until P9, L11. Additional factors may affect the measurement (sway) and derived momentum transfer, like mechanical properties (i.e., elasticity), wind conditions and the distribution of canopy mass.

*Where is an introduction of tree water deficit? The manuscript would benefit from (1) an explicit definition in the introduction and (2) description of the magnitude of typical mass changes resulting from variability in internal tree water content.

*What other mechanical properties vary with tree water deficit? And, what else exerts a temporally varying influence on atmosphere-tree momentum transfer?

*Why is the effect of leaf fall and growth not discussed in the introduction? The effect of leaf state on tree mass has long been a major part of the discussion about accelerometer-based methods for tree monitoring (i.e., Selker et al., 2011, AGU abstract H11G-1155).

*Can section 3.4. discuss potential factors causing interspecific differences in momentum transfer? The relationships in Figure 6 did not appear to explain much interspecific variability. Are there other interspecific differences in leaf type/distribution (maybe simply LAI), or branch size/distribution (maybe simply SAI), or LAI:SAI, stem elasticity, etc?

*What are the potential sources of error in the sway measurements? For example, what is the impact of erratic or sustained winds impeding "free" tree sway on the analysis?

Minor edits:

*P1, L12: Do you mean "Land-atmosphere interactions influence meteorological *processes* and hydrological. . ."

*P1, L18-20 pretty much repeats L14-16. I suggest deleting L18-20.

*P2, L6-9: Can you please break-up the long line of citations. Specifically, I recommend placing each study beside the term indicating which study used, for example, dwarf species? Juvenile crowns? Etc.?

*P2, L26-27: The hypothesis in the introduction appears to be that "momentum transfer . . . [is] in response to water-related stem variations [i.e., water deficit induced mass changes]." But, P9, L10-11 broadens this hypothesis to include "water content or leaf fall." I suggest including a statement like, "water content, leaf fall, or other mass changes" in the original hypothesis.

*P2, L28: Change to "momentum transfer *varies* more than often assumed"

*P3, Table 1: Why was a range of wood density values provided for only one species (S. micranthum) but not for the others? Zanne et al. (2009) provides a range for others. For example: G. glabra (0.32-0.94), E. coriacea (0.70-1.13), D. odorata (0.66-1.09), P. anomala (0.31-0.81), and M. scleriphylla (0.40-0.58). This range shows that, e.g., E. coriacea and D. odorata are not that different in density – despite the table indicating one is "intermediate" and the other is "high" density.

*P3, Table 1 and elsewhere: I believe that "Dypterix" should be "Dipteryx"

*P4, L4: Change to "measure the largest signal *that* can be measured,"

*P4, L12: Please remove references to manuscripts under review. The accelerometer details in one or two of the recently-published papers is surely enough to direct readers.

*Figure 6: In my opinion, the height and diameter relationships, being statistically insignificant, don't merit description in the results. Thus, P11, L12-13 could be deleted.

*P12, L3-4: "leaf flush or fall" is not a "water variation" – sure, water is a part of it, but this statement, as is, is inaccurate.

---

## Author Comment (AC1) · 6 Sep 2018

We would like to thank the reviewers for the constructive feedback. In our revised manuscript, we closely followed the recommendations made by the reviewers. In the following, the comments are listed, with our replies and changed in the manuscript in **bold**.

**Reviewer 1**

The manuscript "Tree-atmosphere interaction responds to water-related stem variations" brings novel data and interpretation on the dynamics of atmosphere forest interaction by applying an original methodology. In short it suggests that changes in tree mass due to lower water content relates to transfer of momentum from air masses to trees, quantified by their sway. The manuscript would benefit by a more careful structuring as is quite repetitive in several instances.

**We thank the reviewer for the constructive comments. We have followed the reviewer's suggestions on restructuring the manuscript and textual changes.**

It did kept me wondering to what extent does tree water content varies. Is there any data out there quantifying water volume/mass changes trough time? In percentage, how much do mass varies in a tree (1%, 5%, 25%)?

**A back-on-the-envelope calculation for a typical tropical rainforest tree, assuming (1) internal tree water content during a day is mainly caused by transpiration (and is not refilled during the day), (2) a canopy surface of 300 – 1000 m$^2$ (Herwitz, 1985) and (3) 5mm of transpiration per day, results in 5000 kg mass variation. For a 50,000 kg tree ( estimated mean above ground biomass of the trees in this study) this would mean a 3 – 10% mass variation. However, this is very speculative and therefore not included in the manuscript.**

In addition, is it possible that a lower water potential on the tree trunk cause changes in elasticity (flexibility) of the trunk?

**We agree with the reviewer and expanded the discussion on this alternative explanation:**

**Page 13, line 6: "An alternative explanation of the changing tree-atmosphere momentum transfer might be the changing elasticity of the tree. A weak relation was found between β and wood density, which can be considered a proxy for tree elasticity. Recent work found a relation between moisture content and elasticity of trees (Mvondo et al., 2017). Although this might be not directly mass related, this would still imply a relation between tree-atmosphere momentum transfer and the tree water status. Also this hypothesis points towards a measureable effect of the transition from wet to dry months on tree-atmosphere momentum transfer."**

Furthermore, due to decreased leaf area index during dry months, is it possible that the changes in wind patterns within canopies space (not measured in this study) produced the observed temporal patterns?

**The reviewer raises a valid point. We expanded the discussion:**

**Page 13, line 23: "Furthermore, this would allow us to investigate whether wind patterns within the canopy have changed over time, which might also explain the variation in tree-atmosphere momentum transfer."**

Because the speculative nature of this study (conclusion depending on an inferred water content), if would be great to have a deeper discussion on other alternative hypothesis.

**The following was added:**

**Page 13, line 6: "An alternative explanation … tree-atmosphere momentum transfer."**

**Page 13, line 23: "Furthermore, this would allow us to investigate whether wind patterns within the canopy have changed over time, which might also explain the variation in tree-atmosphere momentum transfer."**

Page 1: Line 5: Better use monitoring than "measurement".

**Changed.**

Page 1: Line 2: Not sure what "meteorological fluxes" means.

**Changed to:**

**Page 1, line 12: "influence energy, water and momentum fluxes".**

Page 1: Line 7: to "vary considerably" is quite vague. Provide a percentage (or range) of variation. Furthermore, need to specify "variations" in what (I suppose tree trunk diameter)? This also apply to the title of the manuscript.

**Based on the variation of damping β, this has been changed to:**

**Page 1, line 7: "up to a factor of 2.5".**

Page 2: Line 27: no need for capitalized "We". Line 28: change ". . .momentum transfer various more..." to "momentum transfer varies more"

**Changed.**

Page 3; Line 8: Can you indicate the source of the meteorological data?

**Included "and were retrieved from the National Institute of Amazonian Research (INPA)".**

Page 3; Line 11: A bit repetitive (7 tree species).

**Changed to:**

**Page 3, line 11: "A total of 19 individual trees (7 species) were measured during this experiment, covering a broad range of average height and wood density."**

Page 3; Line 15: Change to "measuring tape".

**Changed.**

Page 4; Line 13: How long was the point dendrometers allowed to settle before reliable measurements could be taken. I mention that because it is said that regular band dendrometers requires some months to settle in place, preventing immediate reliable monitoring.

**The dendrometers were installed late July, 2015. In this study we used the data starting on 1 August 2015. The used dendrometers (ZN12-T-2IP, Natkon.ch, Switzerland) do not cause any disturbances by deformations of the outermost layer of the bark, and the spot of measurement is not influenced by the thread rods (Zweifel, 2014). Other studies using similar point dendrometers did not report any uncertainties induced by not allowing for additional settling time (e.g. Zweifel et al., 2010; De Schepper and Steppe, 2010; Etzold et al., 2011; Ehrenberger et al., 2012).**

Page 12; Line 4-5: Higher biomass do not necessarily relates to higher water content.

**We agree. The statement has been rephrased to:**

**Page 10, line 11: "This suggests that changes in β can be explained by mass variations, either caused by for example leaf flush and fall, or water content related biomass changed."**

Page 12; Line 6: "The relation between β and can explain the observations in Fig. 5 and 6A-D." this phrase needs improvement. The observations suggest a relationship.

**Expanded to:**

**Page 11, line 3: "The observations in Fig. 5 and 6A suggest a relationship between β and tree biomass. Mean β and biomass show a clear relationship (Fig. 6H), suggesting that temporal changes in β are related to temporal changes in tree mass. During periods of increased water-related stem variation during the dry months, β decreases for each tree, providing additional observational support for this hypothesis."**

**Reviewer 2**

This manuscript applies a novel, low cost, field-durable method to examine an oft oversimplified process in land surface models: the momentum transfer between the atmosphere and tree canopies. Its major contributions include (1) being the first to achieve in situ measurement of tree-atmosphere momentum transfer, and (2) finding significant fine-scale spatial variability (individual tree) and coarser-scale temporal variability (over weeks and seasons) in tree-atmosphere momentum transfer. Considering the high spatiotemporal variability observed, it would have been nice to see some backof-the-envelope estimate of its relevance to related processes, like those mentioned in the introduction (i.e., transfer of water, heat or gasses). But, such an estimate is not necessary. This method, dataset and analysis would be of high interest to the geoscience community with some revision.

**We thank the reviewer for the constructive feedback.**

Why does the title and abstract focus solely on "water-related stem variations"? The authors discuss the potential effect of leaf mass variations late in the manuscript - leaves are not mentioned until P9, L11. Additional factors may affect the measurement (sway) and derived momentum transfer, like mechanical properties (i.e., elasticity), wind conditions and the distribution of canopy mass.

**We agree that there are several other factors that can influence tree sway. We also discuss these in the introduction and especially the Section 3.4 Synthesis. The main focus point is however the relation between the tree sway measurements and the water deficit measurements obtained from dendrometers. In particular, we believe the simple accelerometer measurements might help in improving the understanding of the response of Amazon trees (and the forest overall) to droughts.**

Where is an introduction of tree water deficit? The manuscript would benefit from (1) an explicit definition in the introduction and (2) description of the magnitude of typical mass changes resulting from variability in internal tree water content.

**We included the following:**

**Page 2, line 17: "We derived a measure for canopy-atmosphere momentum transfer, which are compared to dendrometer based tree water deficit measurements, defined as the difference between a contructed tree growth line and original dendrometer records \citep{Ehrenberger12}."**

**Mentioning typical mass changes for these trees would be very speculative. A back-on-the-envelope calculation for a typical tropical rainforest tree, assuming (1) internal tree water content during a day is mainly caused by transpiration (and is not refilled during the day), (2) a canopy surface of 1000 $m^2$ and (3) 5mm of transpiration per day, results in 5000 kg mass variation. For a 50,000 kg tree this would mean a 10% mass variation.**

What other mechanical properties vary with tree water deficit? And, what else exerts a temporally varying influence on atmosphere-tree momentum transfer?

**The other mechanical property that might change is the tree elasticity. This has been included in the discussion:**

**Page 13, line 6: "An alternative explanation … tree-atmosphere momentum transfer."**

**Other factors influencing tree-atmosphere momentum transfer is the varying wind patterns within the canopy, which is now also discussed:**

**Page 13, line 23: "Furthermore, this would allow us to investigate whether wind patterns within the canopy have changed over time, which might also explain the variation in tree-atmosphere momentum transfer."**

Why is the effect of leaf fall and growth not discussed in the introduction? The effect of leaf state on tree mass has long been a major part of the discussion about accelerometer-based methods for tree monitoring (i.e., Selker et al., 2011, AGU abstract H11G-1155).

**This is now included in the introduction:**

**Page 2, line 13: "Previous work suggested that tree sway data can be used to measure tree mass variations in response to diurnal variations in water content (Llamas et al., 2013), leaf fall or flush (Selker et al., 2011), or intercepted precipitation (van Emmerik et al., 2017)."**

Can section 3.4. discuss potential factors causing interspecific differences in momentum transfer? The relationships in Figure 6 did not appear to explain much interspecific variability. Are there other interspecific differences in leaf type/distribution (maybe simply LAI), or branch size/distribution (maybe simply SAI), or LAI:SAI, stem elasticity, etc?

**We included the following:**

**Page 13, line 12: "Variation between species are combination of multiple factors. From Fig. 6 it is clear that different wood density and AGB can be related to different β. Additional factors influencing the variation between species include tree architecture, leaf type, wind speed within the canopy and stem elasticity."**

What are the potential sources of error in the sway measurements? For example, what is the impact of erratic or sustained winds impeding "free" tree sway on the analysis?

**The following was added to the description of the experimental setup:**

**Page 4, line 12: "This approach assumes that the trees are free standing, which was one of the selection criteria. As this was based on visual inspection, this might be a potential source of error."**

Minor edits:

P1, L12: Do you mean "Land-atmosphere interactions influence meteorological *processes* and hydrological. . ."

**Yes. Changed.**

P1, L18-20 pretty much repeats L14-16. I suggest deleting L18-20.

**Changed.**

P2, L6-9: Can you please break-up the long line of citations. Specifically, I recommend placing each study beside the term indicating which study used, for example, dwarf species? Juvenile crowns? Etc.?

**Changed.**

P2, L26-27: The hypothesis in the introduction appears to be that "momentum transfer . . . [is] in response to water-related stem variations [i.e., water deficit induced mass changes]." But, P9, L10-11 broadens this hypothesis to include "water content or leaf fall." I suggest including a statement like, "water content, leaf fall, or other mass changes" in the original hypothesis.

**Changed.**

P2, L28: Change to "momentum transfer *varies* more than often assumed"

**Changed.**

P3, Table 1: Why was a range of wood density values provided for only one species (S. micranthum) but not for the others? Zanne et al. (2009) provides a range for others. For example: G. glabra (0.32-0.94), E. coriacea (0.70-1.13), D. odorata (0.66-1.09), P. anomala (0.31-0.81), and M. scleriphylla (0.40-0.58). This range shows that, e.g., E. coriacea and D. odorata are not that different in density – despite the table indicating one is "intermediate" and the other is "high" density.

**We thank the reviewer for pointing this out. The table has been updated. Indications (low, intermediate, high) are now omitted.**

P3, Table 1 and elsewhere: I believe that "Dypterix" should be "Dipteryx"

**Changed.**

P4, L4: Change to "measure the largest signal *that* can be measured,"

**Changed.**

P4, L12: Please remove references to manuscripts under review. The accelerometer details in one or two of the recently-published papers is surely enough to direct readers.

**Changed**.

Figure 6: In my opinion, the height and diameter relationships, being statistically insignificant, don't merit description in the results. Thus, P11, L12-13 could be deleted.

**Changed.**

P12, L3-4: "leaf flush or fall" is not a "water variation" – sure, water is a part of it, but this statement, as is, is inaccurate

**Changed to:**

**Page 10, line 11: "This suggests that changes in β could be explained by mass variations, caused by either leaf flush and fall, or water content related biomass changes."**

[revised manuscript text omitted]